# Examples and Results of Aerial Photogrammetry in Archeology with UAV: Geometric Documentation, High Resolution Multispectral Analysis, Models and 3D Printing

**José Ignacio Fiz** [1,2,*], **Pere Manel Martín** [2,3] , **Rosa Cuesta** [4], **Eva Subías** [2], **Dolors Codina** [3] **and Antoni Cartes** [5]

1   Catalan Institute of Classical Archaeology (ICAC), Rovira i Virgili University (URV), 43001 Tarragona, Spain
2   Department of History and Art History, Rovira i Virgili University, 43002 Tarragona, Spain; peremanel.martin@urv.cat (P.M.M.); eva.subias@urv.cat (E.S.)
3   Baula Recerca Arqueològica SL, 17001 Girona, Spain; info@baulaarqueologia.com
4   Unidad de Cultura-Monasterio de San Agustín, Diputación de Burgos, 09002 Burgos, Spain; rcuesta@diputaciondeburgos.es
5   Ajuntament de la Ràpita, 43540 Tarragona, Spain; tcartes@larapita.cat
*   Correspondence: joseignacio.fiz@urv.cat

**Abstract:** The use of unmanned aerial vehicles (UAVs, also known as drones or RPA) in archaeology has expanded significantly over the last twenty years. Improvements in terms of the reliability, size, and manageability of these aircraft have been largely complemented by the high resolution and spectral bands provided by the sensors of the different cameras that can be incorporated into their structure. If we add to this the functionalities and improvements that photogrammetry programs have been experiencing in recent years, we can conclude that there has been a qualitative leap in the possibilities, not only of geometric documentation and in the presentation of the archaeological data, but in the incorporation of non-intrusive high-resolution analytics. The work that we present here gives a sample of the possibilities of both geometric documentation, creation of 3D models, their subsequent printing with different materials, and techniques to finally show a series of analytics from images with NGB (Nir + Green + Blue), Red Edge, and Thermographic cameras applied to various archaeological sites in which our team has been working since 2013, such as Clunia (Peñalba de Castro, Burgos), Puig Rom (Roses), Vilanera (L'Escala, Girona), and Cosa (Ansedonia, Italy). All of them correspond to different chronological periods as well as to varied geographical and morphological environments, which will lead us to propose the search for adequate solutions for each of the environments. In the discussions, we will propose the lines of research to be followed in a project of these characteristics, as well as some results that can already be viewed.

**Keywords:** virtual archaeology; 3D print; UAVs; photogrammetry; remote sensing

## 1. Introduction

### 1.1. The "Democratisation" of the RPA

In the first decade of the 21st century, a new technology that allowed low-level and low-cost flights made its appearance in archaeology: the remotely piloted aircraft (RPA). To this, we added the exponential development of photogrammetry programs, which allowed a qualitative leap in the geometric documentation of archaeological sites.

This technological adoption, or "democratisation" of use, as defined by Poirier et al. [1], has produced a quantitative and qualitative leap in scientific production comparable to what the appearance of PC meant, or the proliferation of geographic information systems (GIS) implemented in free software in the last decade of the last century. Numerous archaeological excavations have been geometrically documented thanks to the use of both RPA and high-performance photogrammetric software, capable of moving large amounts of digital information [2]. According to Themistocleous [3,4], these offer an affordable, reliable,

and simple method of obtaining information on cultural heritage, thus providing a more efficient and sustainable approach to the documentation of structures. Unmanned aerial vehicles have proven highly valuable in the fields of archaeology and cultural heritage, as they provide a non-invasive, time and cost-efficient way to document cultural heritage [5].

For the same reason that the GIS meant an advance in landscape archaeology studies, this "democratisation" of the RPA has caused two changes—not only methodologically, but also in aspects such as the dissemination and socialisation of archaeology. Let us bear in mind, for example, that the application of remote sensing techniques are non-invasive, subject to, and dependent on variables such as platforms (plane, satellite), the resolution of the result and the periodicity of capturing the information, which were not economically accessible to archaeological projects, as well as not very flexible to taking ad-hoc samples, and therefore not the best conditions for optimal data collection to analyse. It should not be forgotten that public projects such as the EU Sentinel program have provided access to high-resolution, low-cost multispectral material for remote work. This material is effective for macro-regional scale or peri-urban studies of large settlements, but limits its effectiveness for micro-regional or small studies.

Finally, it is in the socialisation of the results where the use of RPA acquires its full dimension, the visual possibilities of the 3D models are obtained, the basis for virtual reconstruction is built, and the template from which 3D printing comes into play, opening up room again for "democratisation" of the media to disseminate and socialise knowledge, bringing them closer to groups subject to social exclusion.

In this work, we present a sample of all of the associated activities and results that we have obtained thanks to the use of a UAV throughout an experience starting from 2013, in the archaeological site of Cosa (Ansedonia, Italy), to the present. This knowledge was obtained in the field and in the laboratory from our participation as members of the intervention projects at the archaeological sites (Figure 1) of Puig Rom (Roses, Girona) between 2015 and 2021; Vilanera (L'Escala, Girona) between 2019 and 2021; and in Clunia (Peñalba de Castro, Burgos) in 2021.

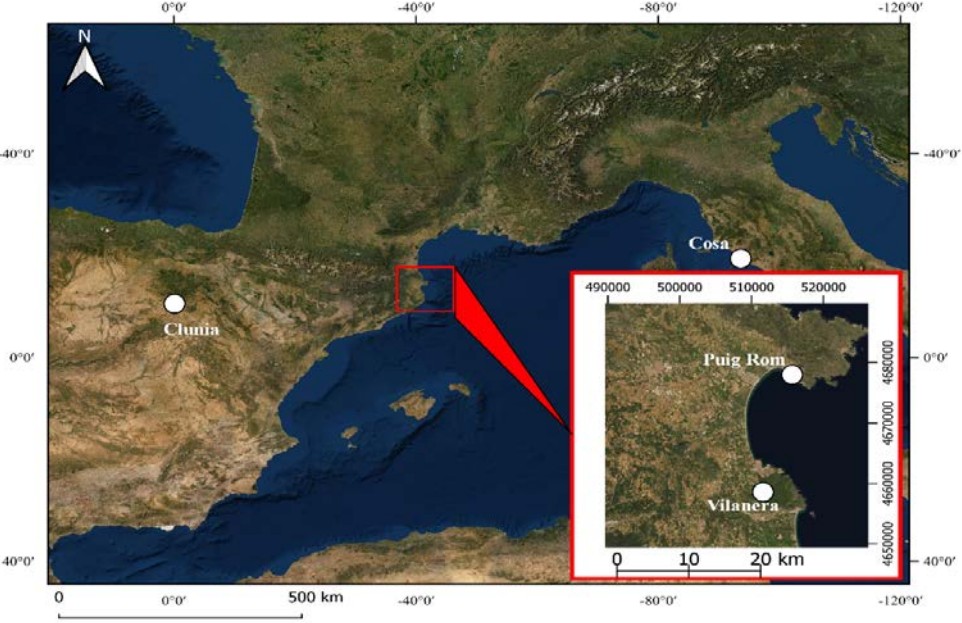

**Figure 1.** Location of the archaeological sites treated in the article. Image source: ESRI World Imagery.

*1.2. Sample of Deposits*

This experience and the first work with the UAV started with the obtaining of a scholarship for visits to research centres by Dr. Ignacio Fiz (2012 AQU BE 100995), carried out at the Università de la Sapienza in Rome, and within the framework of the archaeological excavation campaigns in the Roman colony of Cosa directed by Dr. Mercé Roca,

then-director of the Spanish mission [6]. From then, it was possible to initiate first contact with RPA flights with the use of photogrammetry software. On the one hand, a series of aerial photographs taken of the archaeological site in previous campaigns was worked on, with a camera installed with a gimbal in a hydrogen balloon. On that occasion, our team also used an RPA, a DJI Phantom I, as a platform for capturing photos, and a GoPro Hero 2 camera, without a gimbal, keeping the camera in a fixed overhead position facing the ground. With differential GNSS, control points were taken on the ground from which the orthophotographs obtained from the site were georeferenced, obtaining with the UAV the first 3D models (Figure 2) of some sectors such as the Capitoline temple of Cosa.

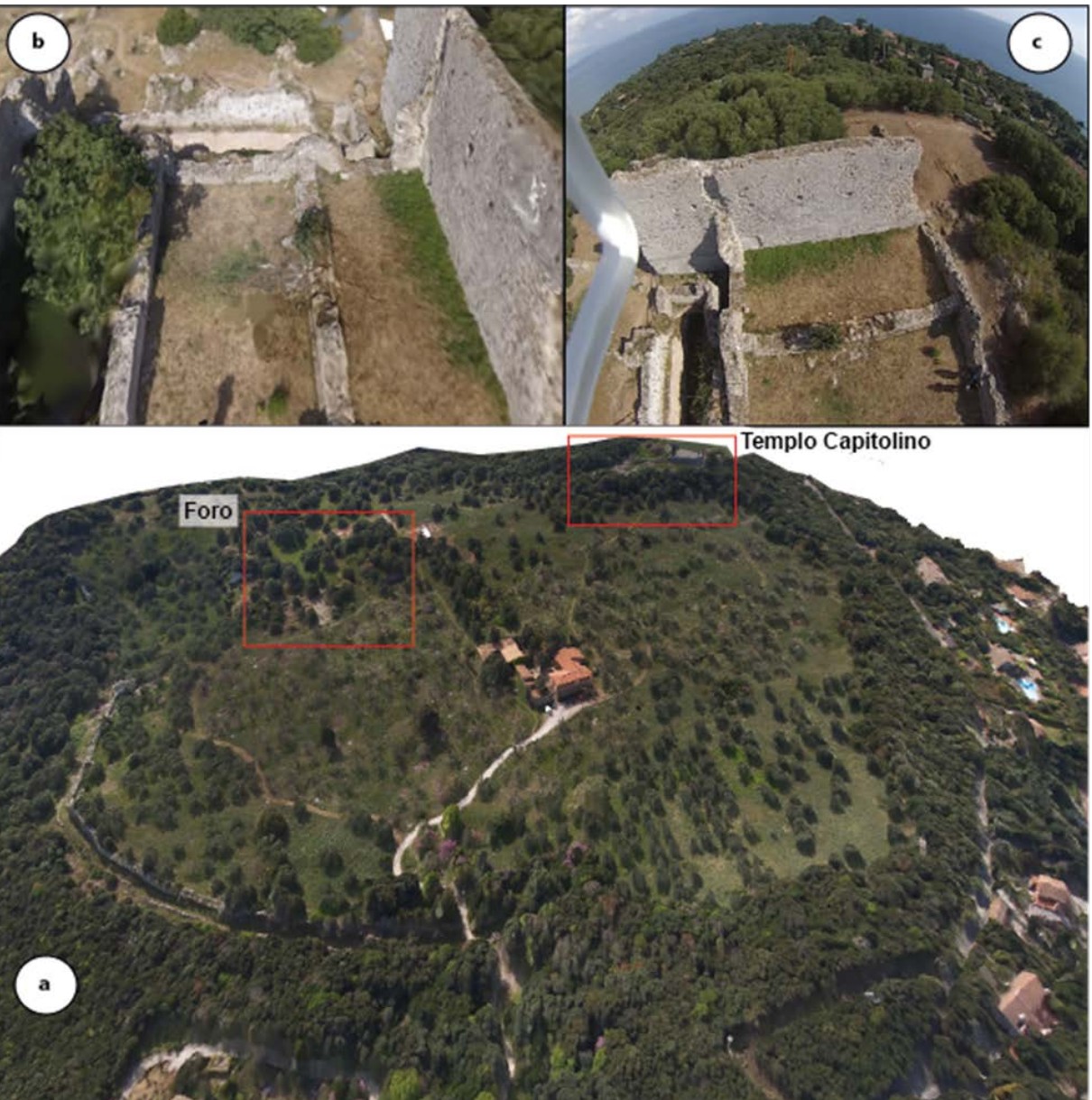

**Figure 2.** The Cosa site (Ansedonia, Italy). (**a**) Textured 3D model, made from aerial photos obtained in 2012 with a hydrogen balloon. (**b**) 3D textured model of the Capitoline temple of Cosa made from a video sequence, taken in 2013, with the GoPro Hero 2 camera installed on a DJI Phantom I UAV. (**c**) Frame of the video captured with the GoPro Hero 2 camera. The lens distortion of this type of camera can be observed. In the first versions of the Agisoft Photoscan program, it was necessary to carry out previous work to correct the distortion.

In 2014, within the framework of the call for four-year research projects of the General Direction of Archives, Libraries, Museums and Heritage of the Generalitat de Catalunya, our team, whose principal investigator was Dr. Eva Subías Pascual, was awarded the project *The fortified nucleus of Puig Rom and its immediate surroundings. Study on the settlement of the Visigoth period in the Sierra de Rodes. VII-X century AD* (no. d'exp. 2014/100582). The town of Puig Rom, or Puig de les Muralles, or the Visigoth castrum (Figure 3), is a paradigmatic monument among the fortifications of this period with a chronology between the 6th and 9th centuries AD. Among the comparable fortifications, it is one of the best preserved and the one with the most careful construction technique. Behind its walled canvas, there is a very important archaeological site from which much can be learned about this period. The works were carried out throughout the 2014–2017 period. For this, it was necessary to have an operator to carry out the flights, since, with the application of the 2017 regulations, it was required that for professional work, an operator must be in charge of taking the pertinent photographs for analysis. This was especially important given that most of the fields described in this article are located in areas that have one or more limitations for the use of RPA, making them impossible for carrying out such flights by a particular user. With the European regulations fully in force as of 2022, the requirements will be different, for which the operator that has carried out the flights in our deposits already has the corresponding accreditation for these new regulatory frameworks and with members of the team among its fleet of pilots.

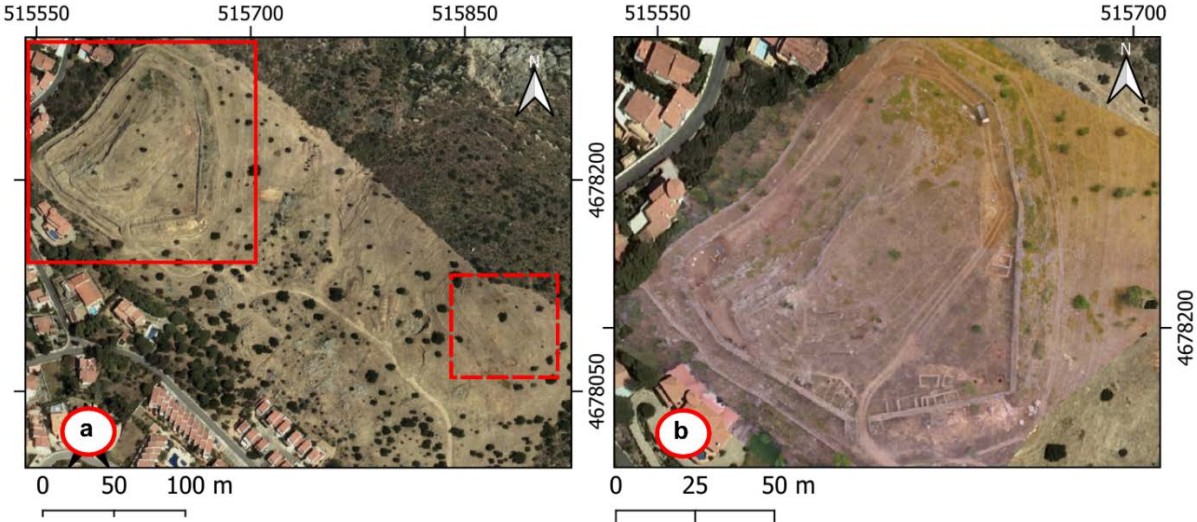

**Figure 3.** Puig Rom site (Roses, Girona). (**a**) Location: box with continuous red line indicates the town of Puig Rom or Puig de les Muralles. The red box with a dashed line indicates the hill near the site analysed and cited in the work. Source: ESRI World Imagery. (**b**) Extension of the Puig Rom walled area in 2017, the result of the first complete high-resolution orthophotography of the site made with UAV flight. ETRS 89 UTM zone 31 coordinates.

At the end of these first methodological tests, one of the final products obtained was a 3D model of the deposit [7].

In 2018, this project was renewed in the next call for four-year plans for the period 2019–2022 with no. of experience 437 K117. In this sense, we have already started working not only with RGB cameras, but also with cameras with NIR bands (from 2019), RedEdge (2020), and thermal bands (2020–2021), with the intention of exploring the possibilities of detecting traces of remains at archaeological sites in areas not only at the site, but also adjoining it. An investigation was started with UAV flights on a hill near the site (Figure 3a. Area delineated by box).

Regarding the Vilanera site (Figure 4), in the same year, 2018, another member of the team, Dr. Dolors Codina, obtained the concession in the same call for the project to excavate

the necropolis de Vilanera (L'Escala, Girona) under the title *The Ancient and Middle Neolithic at the Mouth of the Ter: Archaeological Actions from the Prehistoric Period in Vilanera/Empúries and its Surroundings*, with file number 2018–2021. CLT009/18/00015.

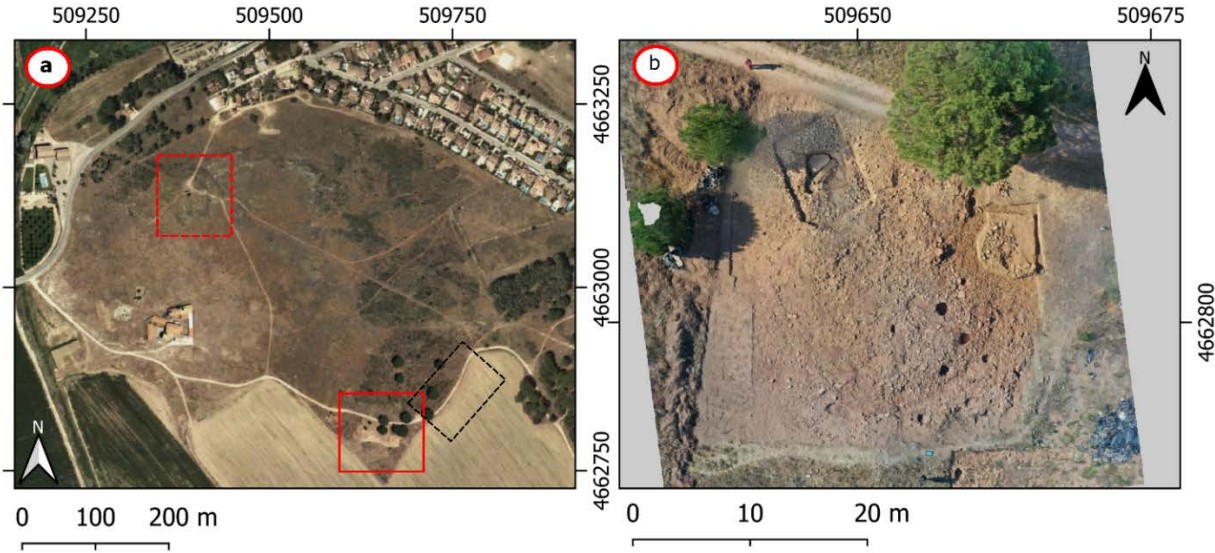

**Figure 4.** Necropolis of Vilanera (L'Escala, Girona). (**a**) General view of where the deposit is located. Black box with dashed line: location of the necropolis. Red box with dashed line: analysed area of the monastery of Sta. Maria de Vilanera. Fountain. ESRI World Imagery. (**b**) Detail of a sector of the necropolis during the 2021 campaign. Flight made with UAV in June 2021. ETRS 89 UTM zone 31 coordinates.

The Vilanera site is the starting point of a project that focuses on expanding knowledge of the establishment and dynamics of the population of the first Middle Neolithic farmers in a coastal territory, in one of the most emblematic areas of the northeastern peninsula—the hinterland of Empúries. The importance of Vilanera is centred on the exceptional nature of the site, located on the edge of a large lacustrine area, at the mouth of the old river Ter, which is flooded with freshwater floods. The excavation work that started in 1999 determined the existence of an immense urn field necropolis, without subsequent alterations. This necropolis, exceptional for its location, its funerary architecture, and its locally produced ceramic materials, is distinguished from its closest parallels by the presence of tombs with imported Phoenician materials, as well as by the presence of bronze elements such as points lance, fibulae, and buckles, typical of the European Bronze Age. Chronologically, this area of the necropolis belongs to the Late Bronze Age–Iron I (900–600 BC).

The 2016 campaign, focusing on the excavation of the central area of the tumulus (Figure 4), provided a large funerary area of more than 40 m in diameter, in the northern part of which a rectangular chamber was drawn, surrounded by stones placed semi-vertically that measured 17 m long by 8 m wide. This exceptional funerary structure, without subsequent alterations, has made it possible, for the first time in the northeast of the peninsula, to excavate a very complex Neolithic tomb where the architectures of the early Neolithic are mixed with those of the middle- and end-Neolithic.

In this sense, apart from the geometric documentation of the site obtained with UAV, we were interested in analysing certain areas near the necropolis. In one of them are the remains of one of the buildings that formed part of the monastery of the Benedictine nuns of Santa Maria de Vilanera (Figure 4a, red box with discontinuous line), founded in 1328. This analysed area is partly at the highest point of the hill of Vilanera, and only conserves a wall of 10 m in length and 4 m at its maximum height. In the environment, it is possible to see the foundations of other structures, but without the possibility of knowing what the structure of the building was like [8].

Finally, in the vicinity of the town of Peñalba de Castro (Burgos) is the Clunia deposit (Figure 5), located on a watch hill called Alto de Castro. Through Sallust (Hist., 2, 93), Tito Livio (Periocas, XCII), Plutarco (Sertorio, 9), or Floro (2.10.9), we know a series of references that tell us about a Celtiberian settlement in the neighbouring hill known as Alto del Cuerno, which had its moment of importance during the Sertorian war. At an indeterminate time between the end of Augustus' mandate, or during that of Tiberius, the foundation of Clunia would have been placed. The city played an important role during the events of the year 68 AD, for which it would receive the epithet of Sulpicia through Emperor Galba and become the capital of Conventus in Tarraconense.

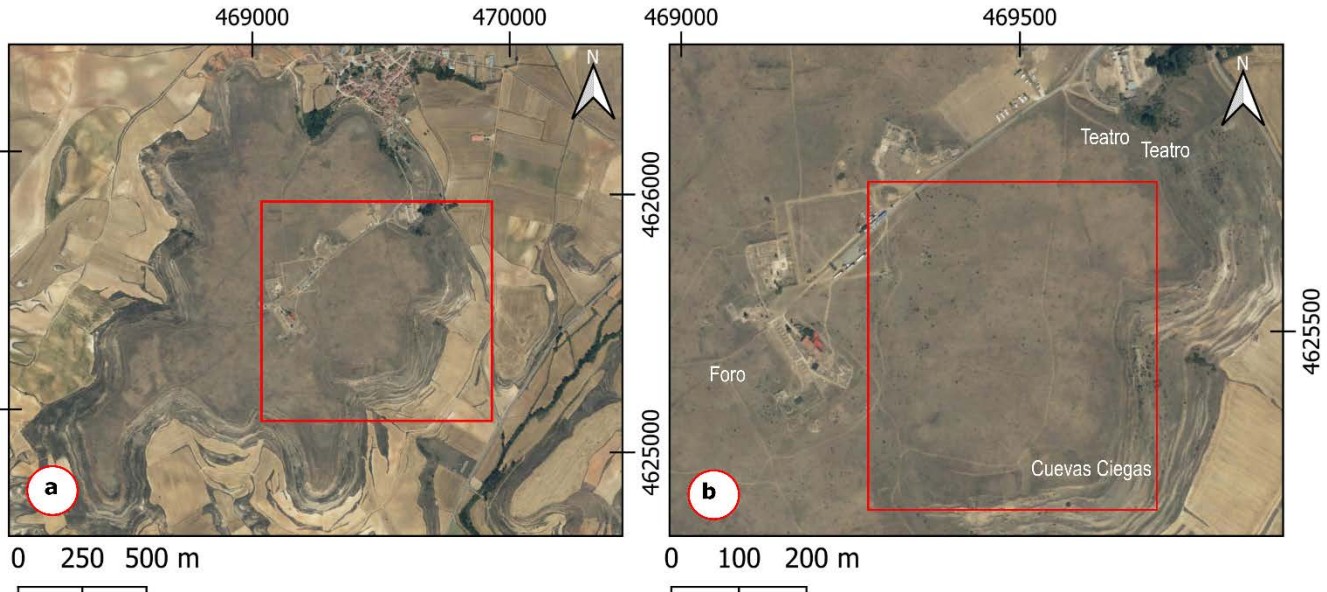

**Figure 5.** Clunia site. (**a**) The Alto del Castro. Red box: analysed sector. (**b**) Detail of the sector analysed between the theatre, the forum, and Cuevas Ciegas. Source: ESRI World Imagery. ETRS89 UTM zone 30 coordinates.

The plateau on which the site sits has an area of 1.2 km², of which only 3% has been excavated since the 19th century. The archaeological site of Clunia is located between the Aranzuelo river (northwest) and the Arandilla (southeast). The hill on which the Roman colony settled offered a series of springs, fountains, or streams, more or less seasonal, which made the place, along with its strategic character, an ideal space for a settlement. The environment of the Clunia deposit, with its karstic characteristics of the hill on which the city sits and which defined its urban characteristics, played an important role in the maintenance and existence of Clunia. There are still many aspects that remain to be analysed, and of which it is impossible for us to know from the surface, such as the urban layout.

Our team in collaboration with the Clunia Research team have tried to analyse an area of the site, delimited between the theatre, the forum, and a group of excavated Roman houses called Cuevas Ciegas (Figure 5b), using techniques of non-invasive remote sensing.

## 2. Materials and Methods

### 2.1. UAVs and Multispectral Cameras

Since our work began in 2014, we have used several UAVs with different characteristics depending on the needs of the project and the available models, as shown in Table 1.

The use of multispectral cameras, such as MAPIR, allows us to configure various light collection filters from a single lens. In the case of our work, we chose to use the NGB (Nir + Green + Blue) and Red Edge (RE) versions, since they share the same general characteristics that we detail (Figure 6).

**Table 1.** List of UAVS and cameras used.

| Year | Aircraft | Camera |
|---|---|---|
| 2014 | DJI Phantom 1 DJI Phantom 2 | GoPro Hero 2 |
| 2016 | | Own camera |
| – | DJI Phanom 3 Advanced | MAPIR NGB |
| 2020 | | MAPIR RE |
| 2018 | Yuneec Thyphoon H | CGO-ET |
| 2019 | Parrot Anafi Thermal | FLIR Lepton 3.5 microbolometer Hasselblad 1″ |
| 2020 | DJI Mavic 2 Pro | MAPIR NGB MAPIR RE |
| 2021 | DJI Mini 2 | Own camera |

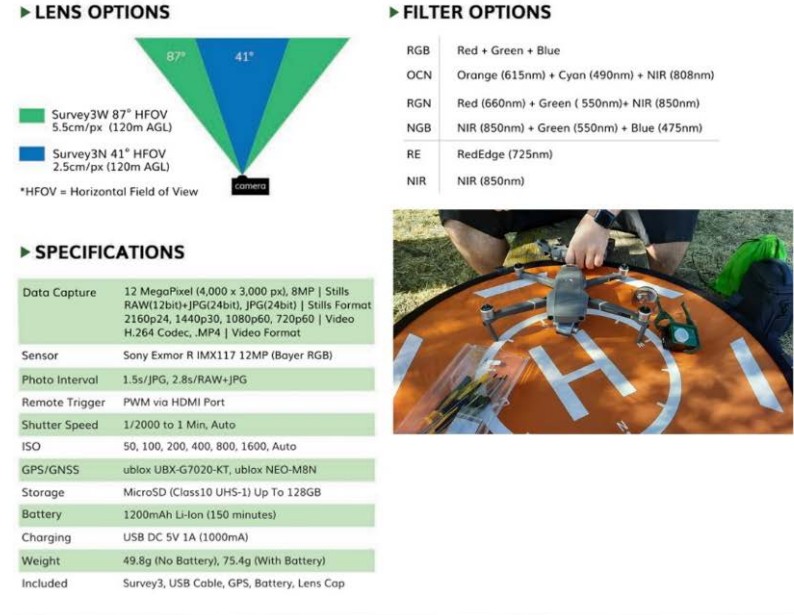

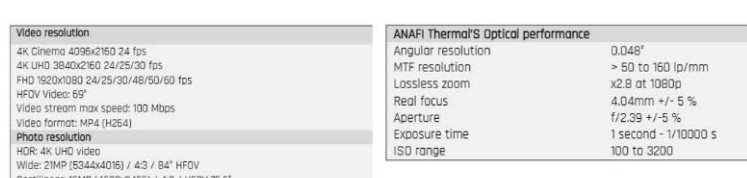

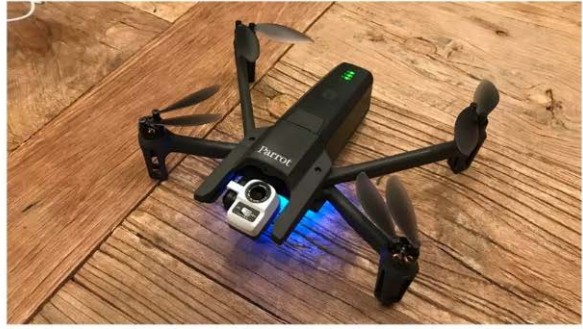

**Figure 6.** Specifications of the cameras. Above: MAPIR system; below: Parrot Anafi Thermal with the FLIR default thermal camera. Extracted from the official Guide Setup provided with the aircraft.

To capture thermal images, we initially used the Yuneec Typhoon H aircraft that had a CGET thermal camera attached, but it was not sufficiently satisfactory during the tests carried out, since the instruments did not collect radiometric information for editing. Given these drawbacks, we opted for the combination of the Parrot Anafi Thermal aircraft (Figure 6), which already contains the built-in thermal camera and with a good resolution, in addition to recording the radiometric data for the correct subsequent treatment with a processing software and generation of thermal indices with correctly calibrated temperatures.

### 2.2. Treatment and Analysis Software

For flight planning, we used the Pix4D Capture app, given its compatibility with the aircraft we used. It includes specific flight profiles for DJI Phantom 3, Dji Mavic 2 Pro, and also for the Parrot Anafi Thermal. In all of our flights, the overlap of the images was always close to 80%, both along and across the flight paths. This allowed us to generate a greater number of photographs to increase the quality of our final models.

To carry out the orthophotograph products with the multispectral cameras attached to our main aircraft, it was recommended that the application values be kept as low as possible in terms of movement speed. Pix4D Capture allowed us to set two data collection modes: safe mode and fast mode, available only for compatible DJI drones. This property is very useful to minimize the risk that the images captured directly with multispectral cameras, which must be taken automatically at regular intervals, are taken at an incorrect angle due to the sudden braking of the RPA. This is done in safe mode, where for each point that we have calculated for taking images, the drone has to stop, take the picture, and start moving again to reach the next point. Therefore, keeping the speed low (slow or normal), and using an appropriate shutter speed on both cameras, will give correct and distortion-free results. This same method can be used for obtaining thermal images, but with some variation in the speed of the same aircraft, whose limit is much lower.

With the various RGB, NGB, RE, and Thermal cameras that we used, and depending on the type of band, we used the photogrammetry software that most facilitated the work of generating orthophotographs. MAPIR cameras require prior processing work (TIFF format generation from the original RAW format) and camera calibration. This process is done using the company's own free program, MAPIR Camera Control.

To do this, we needed some shots taken on the calibration card included with the camera, which allowed us to establish the correct reflectance values for the generation of the .tif files necessary for processing in the Structure from Motion software (Figure 7).

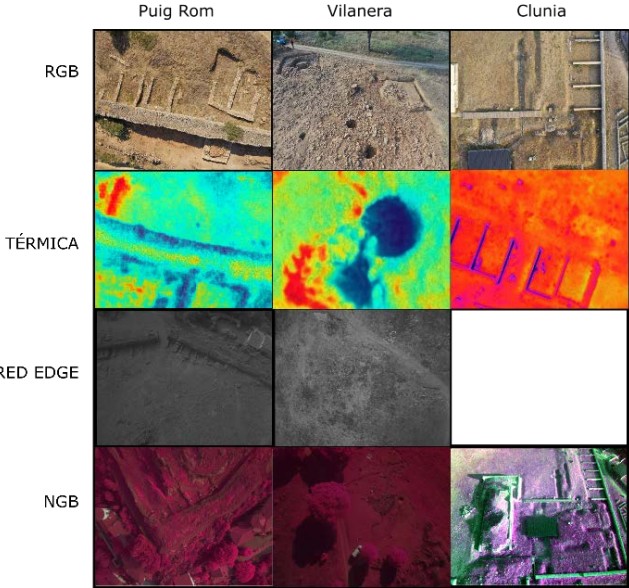

**Figure 7.** Example of unprocessed images for the analysed sites in the text.

The usual photogrammetric treatment with an RGB, NGB, and RE camera was carried out with the commercial program Agisoft Metashape v1.7.4. The program allows the usual procedure of camera alignment, dense cloud, mesh, texture, and orthorectification of images with MAPIR cameras, generating models and derived products with texture in false colour NGB and RE. However, for the treatment of the images obtained with the FLIR thermal camera of the UAV Parrot Anafi Thermal, it is more advisable to do it with the commercial program Pix4D mapper, in its version 4.6.4 or higher. The reason is that the program can and must execute the processing options with the data obtained from the high-resolution RGB camera that is incorporated, and then superimpose the values of the thermal photographs as a texture on the resulting geometry. In this way, we obtained a precise and detailed mesh with the possibility of alternating between two types of texturing. This gave us the possibility of generating a high resolution orthophoto of the thermal band (thermal index).

Among the products generated from the 3D models, we focused our work on obtaining orthophotographs and digital elevation models. With the first ones, the visualization of georeferenced results, image analytics, and index calculations from the bands were carried out using the free software GIS QGIS (QGIS v.310.13-A Coruña) and GIS SAGA v7.9.0 (software for Automated Geoscientific Analyses).

With the tool "band calculator" in the Agisoft Metashape software, we obtained different indices through the superposition of the different bands, and finally combined them in a unique image file. Vegetation indices are effective methods for detecting and quantifying plant health through remote sensing [9]. In reference to archaeology, it should be noted that these indices are useful for the detection of buried structures that alter the growth and phenology of the overlying vegetation [10]. Its use in non-invasive archaeological prospecting has been and continues to be widely accepted, and has good results through satellite images first, and UAVs equipped with multispectral cameras later. For this reason, we used the vegetation indices that involve the set of bands that we are using as analytical techniques, that is, NIR, RE, R, G, and B. It should be noted that by not having a single multispectral camera, but separate cameras, forced us to carry out the additional work of image composition, previously georeferenced with high precision, through control points taken with the GNSS. Among the indices that we used the most are the NDVI (NIR and R), ENDVI (NIR, G, and B) and ARVI (NIR, R, B). Through the combination of the NIR, R, G, and B bands, finally, a principal component analysis (PCA) was applied, especially due to the good results obtained by our team in the Egyptian oxyrhynchite landscape with WorldView 2 satellite images [11] and in the same Clunia project with SPOT and TripleSat multispectral images [12,13].

PCA is a multivariate statistical method that reduces redundant information and the dimensionality of the data, in such a way that the resulting components reflect the maximum variance of the original data set [14]. Aqdus et al. [15] estimated PCA as the most effective visualisation tool for multi and hyperspectral images. For the calculation of the PCA, the SAGA software was used.

With the second ones, we proceeded to the analytics carried out with the RVT Relief Visualization Tools program v2.2.1 (RVT). This program, specially designed for LAS LiDAR classified files, allows the improvement in the presentation of digital terrain models. The program, developed by the research group of the Slovenian Academy of Sciences and Arts [16], has been used with great success in studies by Massini et al. [17] to locate the structures of a medieval fortification in Basilicata (Italy); Garcia Sanchez [18] in the detection of Samniticus forts in Civitella (Longano, Italy), and Roman et al. [19] in the record of archaeological traces of the Roman limes in Dacia at Porolissum. Our team has previously worked with this program, mainly based on LiDAR data provided by the IGN, applying it above all to the Clunia deposit [12,13].

Generally, the highlighting functions of the RVT program have been applied to the digital terrain models obtained from LiDAR data, whose point clouds are classified, in such a way that it is possible to select those that correspond only to the surface, with

the rest being disposable. However, the exceptional improvement of photogrammetry programs, such as the aforementioned Agisoft Metashape, allows the classification of dense point clouds. Works like those of Klápště et al. [20] and Jimenez et al. [21] already focus on comparisons between various software that allows the classification of point clouds obtained from images captured with a UAV.

The second study is interesting, as it analyses the classification based on aspects such as UAV platforms and cameras, flight planning, image acquisition (height, overlap, speed, flight line orientation, camera configuration, etc.), georeferencing, geomorphology, and different types of land uses. This last aspect is of special interest, since it is necessary to remember, as these authors do, that the passive nature of the sensor, in these cases the cameras used, cannot penetrate the vegetation as LiDAR does. In complex situations of vegetation, in which we will find high percentages of surface below vegetation or buildings, it is evident that it will not be possible to obtain a high quality DTM, nor to expect that the RVT would improve the results. Fortunately, the examples that we present correspond to sites in which the vegetation does not present large percentages, as it is very dispersed, or on the contrary, very concentrated in very specific areas in the treated territories.

### 2.3. 3D Editing and Printing Software

For the treatment of the three-dimensional models obtained through some of the software described in the previous section, we resorted to the free software Blender for the manipulation of the models, in .obj format, and thus obtained a much more orderly geometry without topology errors so as to be able to make 3D prints of some parts of the deposits. For the 3D printing, we removed all kinds of material or generated texture, as it is essential to keep the geometry as detailed as possible. That is why, in the previous step, when moving from the dense point cloud to the mesh, we needed to increase the number of faces in order to have a surface that is much more adjusted to reality, and that is easily interpretable by the different printing formats. The most used are filament deposition (FDM) and resin printing by applying UV light to LCD screens (photocuring).

The management of the models obtained was carried out using the free Cura 4.10 software for filament printers, and the ChituBox for prints generated with resin. Both impressions are based on different technologies, which we analysed in their corresponding point.

To make the 3D filament prints (PLA), two Creality brand machines were used, specifically the Ender 3 Pro and the Ender 3 Pro V2. Each were made on a bed of tempered glass, and with a layer height of no more than 0.16 mm from a 0.4 mm nozzle.

These square-sized printers allowed us to make approximate prints that did not usually exceed 20 cm in width and length, as well as 25 cm in height.

For the resin, the printer used was an Elegoo Mars, whose reduced dimensions allowed us to obtain a maximum of 15 cm on its z axis, 12 cm on its Y axis, and 8.5 cm on its X axis.

## 3. Results

### 3.1. 3D Models

One of the first results obtained once the captured images were processed was the creation of 3D models (Figure 8) to generate the orthophotographs that would allow us to work on the delineation of the structures of architectural walls, and generate all of the plans required by the administration. To complement these orthophotos, control points or CGPs were added that were taken with an Emlid Reach + RTK GPS; in this way, the geolocation of the material was more precise and therefore more reliable to be presented as a scientific result.

This last point was of such importance that the fact of not having the centimetric precision that control points can offer us would invalidate many of the hypotheses that we proposed through the study of the results obtained with the aforementioned multispectral cameras. Although the product presented is a raster image in 2D format, the ability to obtain photographs from various angles and heights gave us the chance to generate a complete three-dimensional model of the treated sites, especially Vilanera and Puig Rom, which have visible structures. The generation of the models was carried out with the software Agisoft

Metashape for Puig Rom, and Reality Capture to generate the three-dimensional model of Vilanera. Both works allowed us to evaluate the potential of each of the models and in which situations it is better to use them.

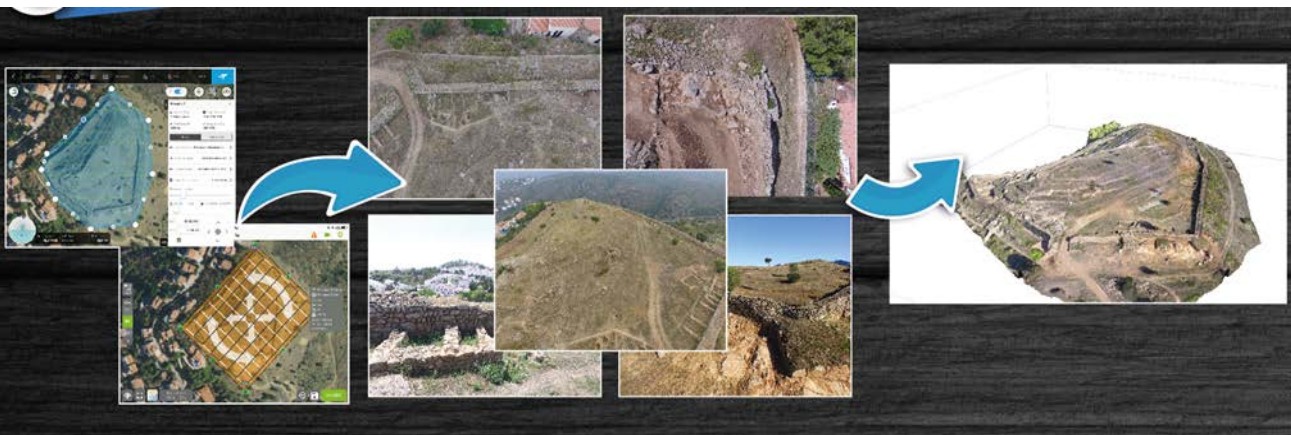

**Figure 8.** Scheme of geometric documentation through the use of drones. On the left, we can see the interfaces of the apps used to plan the flights. In the centre are the images obtained, and on the right, the resulting 3D model on which we carried out the tests and analysis.

A separate case deserves the treatment of the 3D models when we used the data from the thermal cameras, since, as they contain radiometric information, they have to be analysed with Pix4D, as this software is specially optimised for this type of information, thus giving results such as those presented in this work.

Some of the first results of geometric documentation were presented in 2018 [5] in congresses and conferences dedicated especially to the dissemination and enhancement of geometric documentation tasks, with their problems and solutions. For this, the Blender software was used with the generated 3D model, which was then rendered to create an audio-visual piece that allowed us to see the current state of the site virtually. These works were part of a project aimed at the dissemination and signage of heritage elements (which is still in process). The possibility of adding environmental effects such as lights, clouds, and depth of field helped us make the said visualisation more understandable to the non-specialist public, and to better understand the scope of the site in its first current excavation phases (Figure 9). However, the fact of being able to generate three-dimensional models with a high number of faces—more than 40 million—allowed us to be able to make a first recognition of those structures and of the different anomalous shapes that part of the interior of the site presented.

Combined with the DTM generated for the studies with a GIS, the study perfectly complemented us, since we had an orthogonal view in two dimensions that corresponded to its three-dimensional replica. It was also very helpful to be able to work without a distracting texture during surface analysis. For this reason, on some occasions, we used the .obj file generated by photogrammetry software without an associated materials and texture file, keeping only the geometry.

These first three-dimensional models allowed us to document and store the information to be used later in making virtual proposals for a reconstruction, as well as being able to locate the site in its immediate surroundings, and carry out different spatial analyses. With the completion of the excavation and documentation project close at hand, this is the moment when all of the obtained data must be put on the table in order to plan these new actions in the field of virtual reconstruction, validate the hypotheses proposed in the different analyses, and check that it is possible that the anomalies presented really coincide with the structural elements found under the earth's surface.

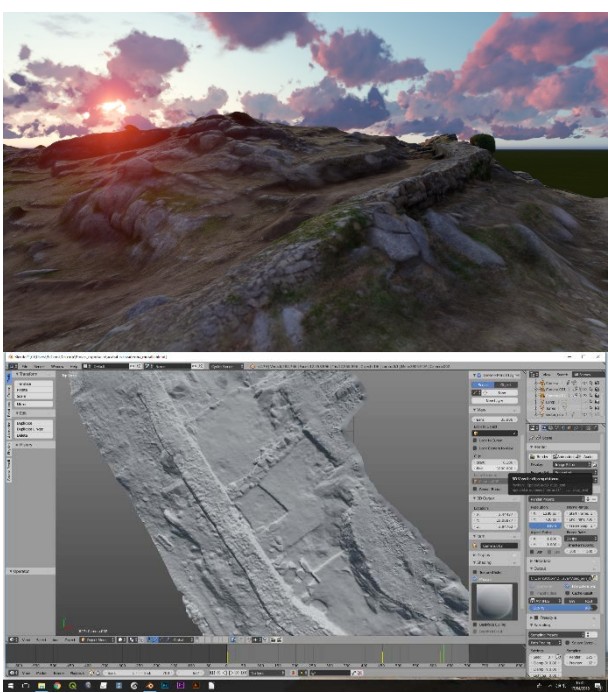

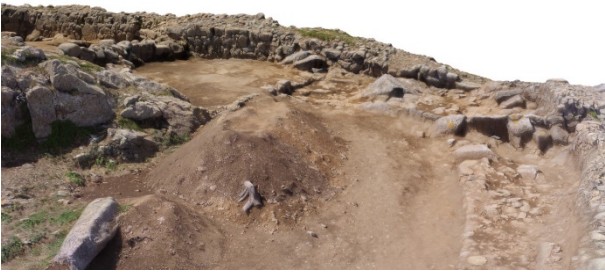

**Figure 9.** At the top, we see a frame of the rendered animation. In the central part, we can see part of an excavated sector without texture, suitable for structural analysis. At the bottom is a screenshot of the textured 3D model of the same sector.

### 3.2. Analytics

From the 3D models, we were able to generate orthophotographs, or digital terrain models (DTM), from which we proceeded to carry out the analyses using the methods mentioned in the methodologies section. In this sense, the surface indicator detection technique used since the 1970s and 1980s highlights the presence of archaeological remains, and they are divided into soil marks, crop marks, and microreliefs.

In these cases, we observed very slightly visible indicators in the resulting RGB images, highlighted through the use of other bands such as NIR or Red Edge, or by applying analytics such as PCA or the application of vegetation indices. The microreliefs can be detected thanks to the work with a high-resolution DTM, carrying out an enhancement process through the functions of the RVT program.

Thus, for example, in Vilanera, in two of the flights carried out in February and April 2019, it was possible to carry out not only the geometric documentation and the 3D models of the necropolis, but also, in an area adjacent to the site, by contrasting the NIRGB image and the ENVDI and NVDI indices (Figure 10a–c); traces that could be interpreted as structures were observed (Figure 10d). It should be noted that we observed the presence of ceramic material on the surface in the area where these marks appeared. However, these traces are confused by the presence of others that can clearly be associated with the traces left by the work carried out by a tractor in the field.

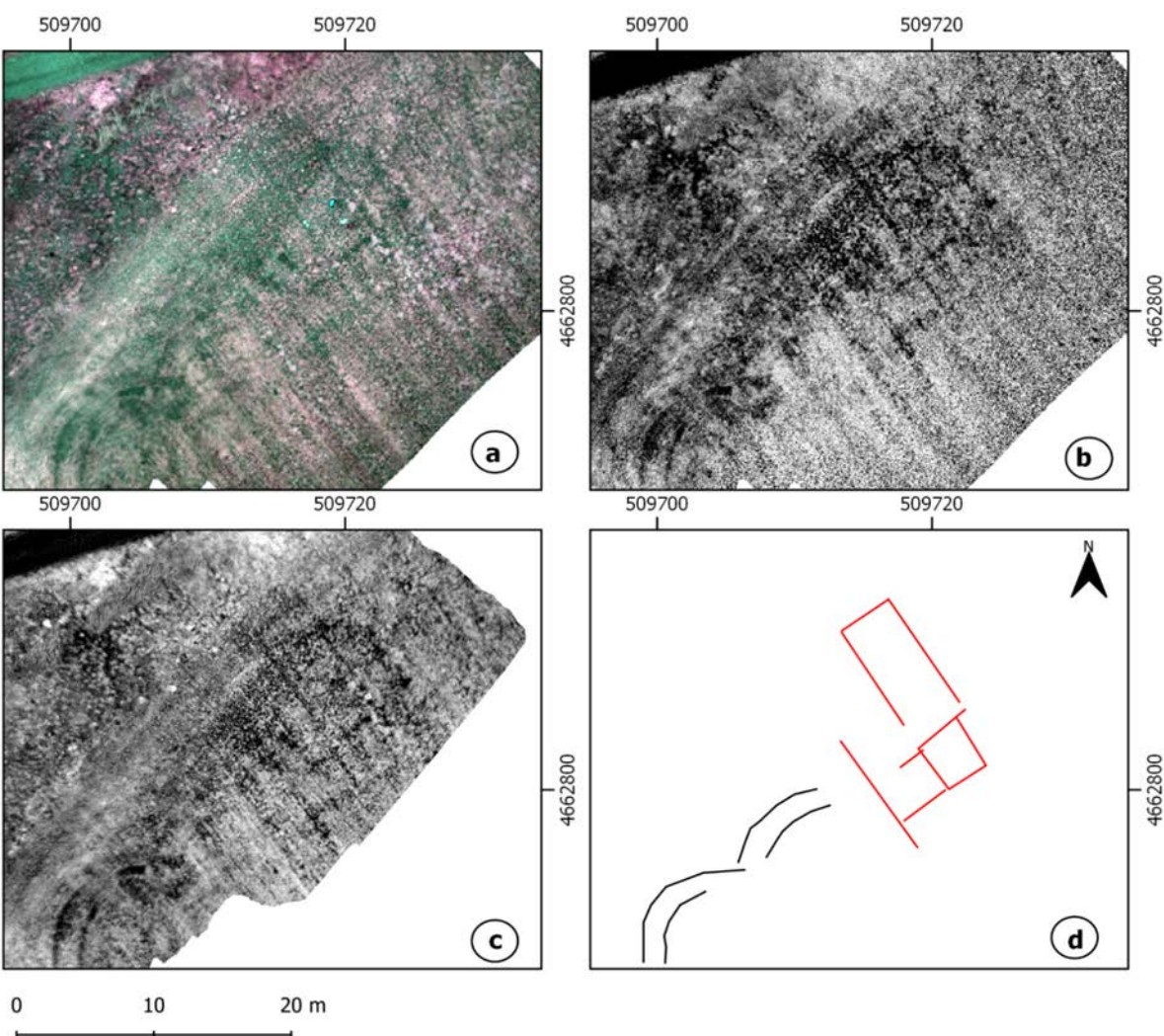

**Figure 10.** Traces detected in the Vilanera deposit: (**a**) NIRGB image; (**b**) image resulting from applying the ENVDI index; (**c**) image resulting from applying the NVDI index; (**d**) interpretation of the traces. Red: structures. Black: traces left by field work. ETRS89 UTM 31.

Another zone of the Vilanera site we worked on is the one corresponding to the monastery of Sta. Maria de Vilanera.

The UAV flights took place in March 2020 and in October 2021.

In this case, we obtained the best results by exploring the microreliefs through the analytics carried out with the RVT program, enhancing the results of the DTM generated from the RGB images captured on the UAV flight in March 2020. In March 2020, the NGB camera was not used, but the camera with the Red Edge band was tested. This time, we calculated the ENVDI index, changing the NIR component of the formula for the Red Edge band.

The results (Figure 11) are certainly interesting, since it is possible to see that both methods combined provided a visual representation of the complete structure of the building, made up of various rooms or spaces that reach a dimension of approximately 15 m wide by 30 m in length.

Another of the examples that we present corresponds to the Puig Rom deposit. Specifically, the analysis was carried out on a hill located just 300 m from the centre of the Visigoth walled enclosure (Figure 12), and at an elevation level (183.46 m above sea level (asl)) that is barely 6 m below its highest level (189.49 m asl), and almost 8 m above the lowest point (174 m asl).

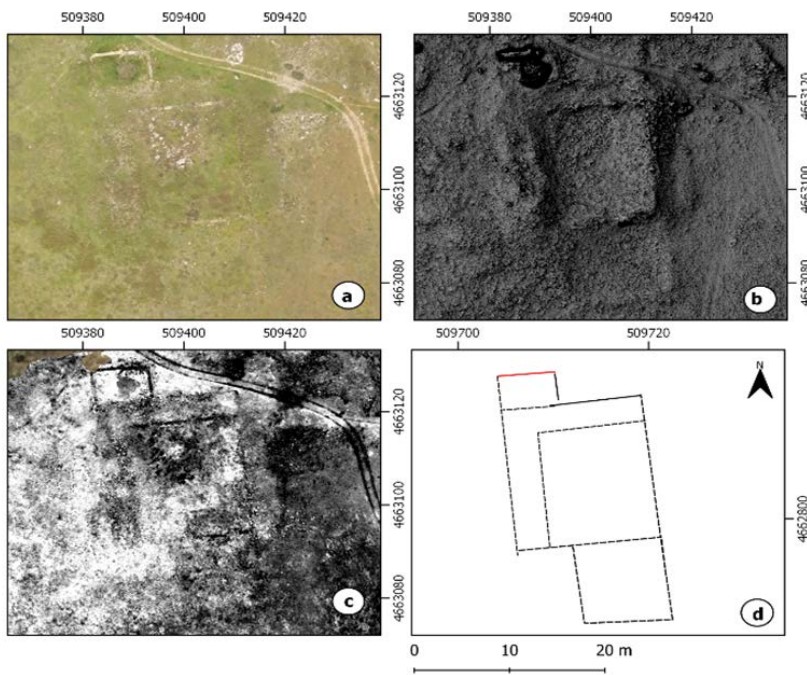

**Figure 11.** Structures of the monastery of Santa Maria de Vilanera. (**a**) RGB image; (**b**) treatment of the DTM with the RVT program, SkyView Factor function; (**c**) ENVDI index to the RedEdge, G, and B bands; (**d**) interpretation. Line in red: wall still existing; black line: foundations; dashed black line: interpretation of the visible traces in images (**b**,**c**). ETRS89 UTM 31.

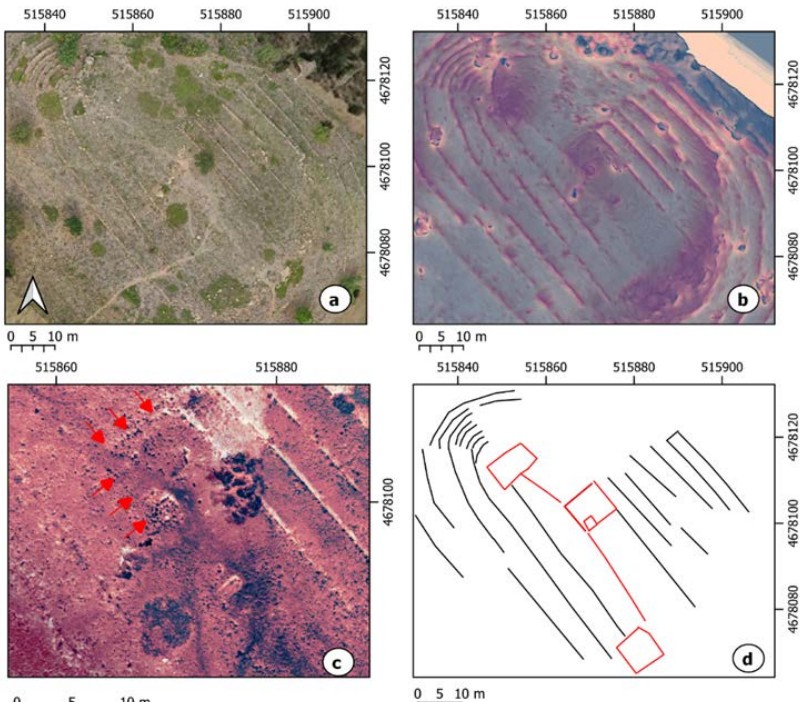

**Figure 12.** Hill attached to the Puig Rom site (Roses, Girona). (**a**) RGB image, October 2021; (**b**) combination of images resulting from the Openness Positive (40% opacity), and Openness Negative (100% opacity) treatments; (**c**) zoom in of the central sector where the traces of a rectangular structure are observed, depicted by the red arrows, showing a PCA result of the NIR, Red Edge, R, G, and B bands; (**d**) interpretation. Red lines: traces of microrelief detected; black lines: agricultural terraces.

Our interest in this hill led us to survey it in 2017 without obtaining any results, and to make it one of the objectives of the RPA flights. Several flights with RGB and NGB bands were carried out in July 2019 and July 2020. In these flights, we obtained the best results through the study of the microreliefs, which, as shown in Figure 12d, revealed the presence of three square shapes. This prompted us to fly in October 2021 with NGB, RGB, and Red Edge bands to improve the conditions in which the indices can be most revealing. In this way, the central rectangular shape, also located at the highest point of the reservoir, was also identified in the result of a PCA analysis from the NIR, Red Edge, R, G, and B bands.

The last of the results that we present in this work come from the study with RPA in an area of the Clunia deposit. The information collected in the campaign of flights carried out at the end of July 2021 presented various problems including, firstly, the period chosen. The summer season meant that the vegetation indices did not produce appreciable results, and therefore it was necessary to use other analyses described in the methodology section.

Secondly, we encountered a problem we faced previously when the objects of analysis were multispectral images and SAR-X radar taken from satellite platforms. This problem was the high complexity of the analysed extension, highly affected by the parcel subdivision system in use until the involvement of Alto del Castro as a protected archaeological site (Figure 13b). This parcel division, hidden by the passage of time, and revealed in the analyses carried out, makes it difficult to detect traces corresponding to archaeological structures.

However, as can be seen in the images of the various treatments (Figure 13c–e) and in the interpretation made (Figure 13f), the appearance of transverse marks at the orientations corresponding to divisions subdivided gave the impression of archaeological structures from which the subdividing walls were supported.

### 3.3. 3D Printing

One of the purposes of this article is to analyse to what extent the physical three-dimensional representation of the analysed elements can be useful. In this case, four prints were made from two of the sites studied: Vilanera and Puig Rom.

For the first print, the excavation area of the late 2020 campaign was printed, in which the remains of a burial mound appear, as we can see in Figure 14. For the tests using filament printing, the entire excavated area was chosen, since we had the capability to make larger printouts. This initially led us to raise the possibility that the detail would not be high enough for us to be able to distinguish between the various elements found in the chosen section.

The dimensions of the current printed piece were $18.5 \times 22 \times 1.6$ cm, and with a layer height of 0.16 mm, the execution time was 48 h. The selected material was a red PLA from the Smartfils brand, whose post-printing results are good in terms of consistency and visibility of the layers. The extrusion values used for this case were 200–210 °C, at a print speed between 42 mm/s and 50 mm/s, with retraction values of 6–7 mm distance at a speed of 40 mm/s. Given its flat shape, no additional supports were necessary to guarantee the correct printing of areas with overhangs, as well as to obtain correct adhesion to the printing bed and avoid the warping effect that can be generated, particularly in the corners of large dimension models when the bed is not properly levelled and the first layers do not adhere properly. To do this, apart from calibrating the corner springs as efficiently as possible, fixing products were used to increase this adherence. Given the nature of this type of printing, and the fact that it does not have supports, post-processing was not necessary as it is usually. Once the part was removed from the print bed, it was ready to be shown or to make some improvements, such as giving it colour.

At the same time, it was decided to make a much smaller section so it could be printed using a resin printer, since its operation differs significantly from what we had been using up to that moment, such as filament deposition (FDM).

As in 3D deposition printers, the material is deposited at a high temperature by an extruder that has the ability to move the filament in thin layers in one or two axes. This means that the printing times are much longer, as the seconds and minutes that the printer takes to make the paths to deposit the necessary filament at the specific points must be added to the deposition time.

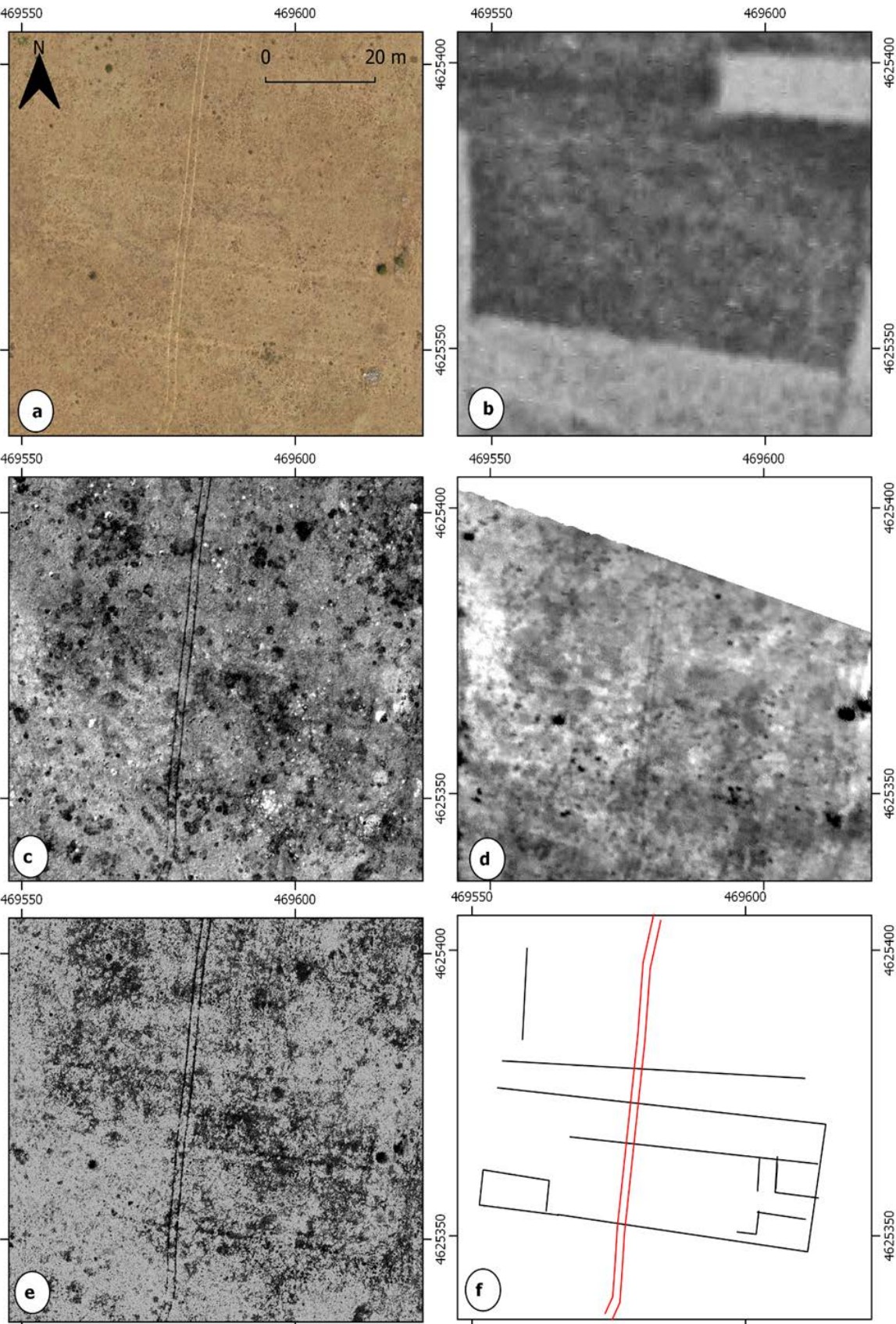

**Figure 13.** (**a**) RGB image; (**b**) USAF flight of 1956; (**c**) NIR band. Flight July 2021; (**d**) Thermal Band. Flight July 2021; (**e**) component 2 PCA bands NIR, R, G, B. Flight July 2021; (**f**) interpretation. Red: current path. Black: traces observed in the results. ETRS89 UTM 30.

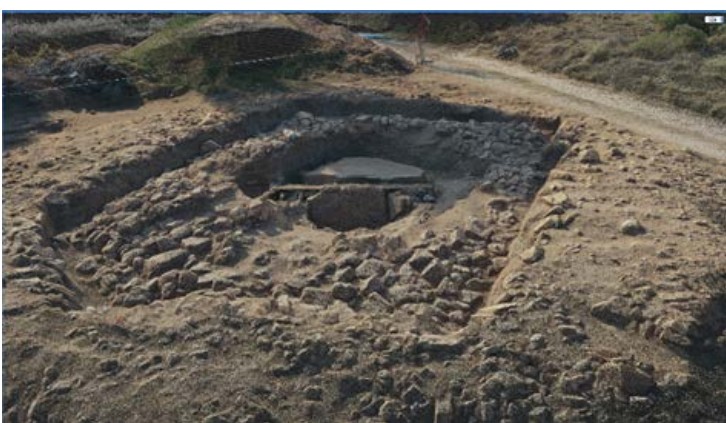

**Figure 14.** Point cloud generated with Reality Capture for 3D printing.

With LCD printers, the layer conception is very different from an FDM printer. In this type of printer, the entire layer is printed at the same time. For this purpose, a UV light source is projected onto an LCD screen that draws the shape of the layer so that it is solidified in a homogeneous way. Depending on the power of the light source and the resolution and quality of the LCD screen (it can be colour or monochrome) and the resin used, it will take more seconds to obtain a solid layer that will adhere to the printing platform. This platform hangs from a z axis that goes up depending on the layer height that we establish; we made prints with a height of 0.05 mm and an exposure time (the seconds that the UV light source is active to solidify the resin) of 10 s. This last value will heavily depend on the type of resin used, its colour, and the combination of light source and screen. The printing time for this piece ranged from 3:15 to 4 h. The size did not exceed the width of the platform, since at this point, we were interested in having a size that matched in scale with the previous impression, so as to be able to see the comparative results of that specific section. Keeping in mind the difference between the two printing methods, supports were required when we printed with resin. If the supports are insufficient, or the space between them is not close enough, errors can occur. The typical errors occurred when lifting the cured resin from the tank, and it may not have adhered correctly, producing a small deformation due to warping (Figure 15b).

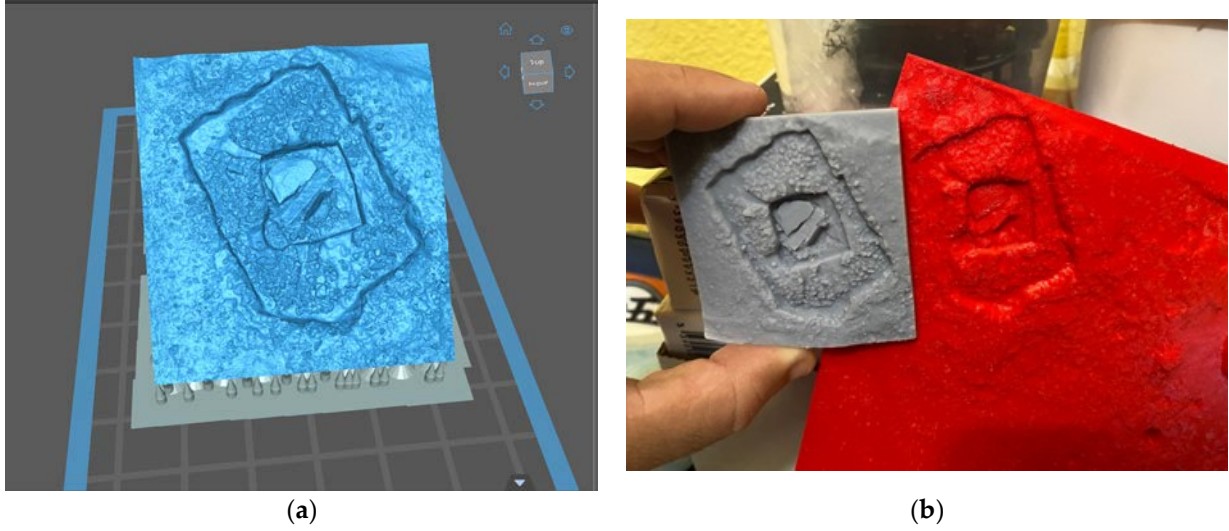

(**a**)　　　　　　　　　　　　　　　　　　　　　　　　　　　(**b**)

**Figure 15.** Printouts of the area belonging to the Vilanera site; (**a**) placement of the piece in the Chitubox rolling mill; (**b**) comparison between the two impressions made: on the left in resin (grey), on the right in PLA (red). The small deformation in the lower left part can be observed in the resin piece.

Finally, the largest printed part was made by completely changing the orientation, and starting from a narrower base so that the suction force when lifting the layer of cured resin was as low as possible, and thus prevent the supports from bending or breaking, because the uncured resin is still very malleable and lacks the final rigidity. (Figure 16).

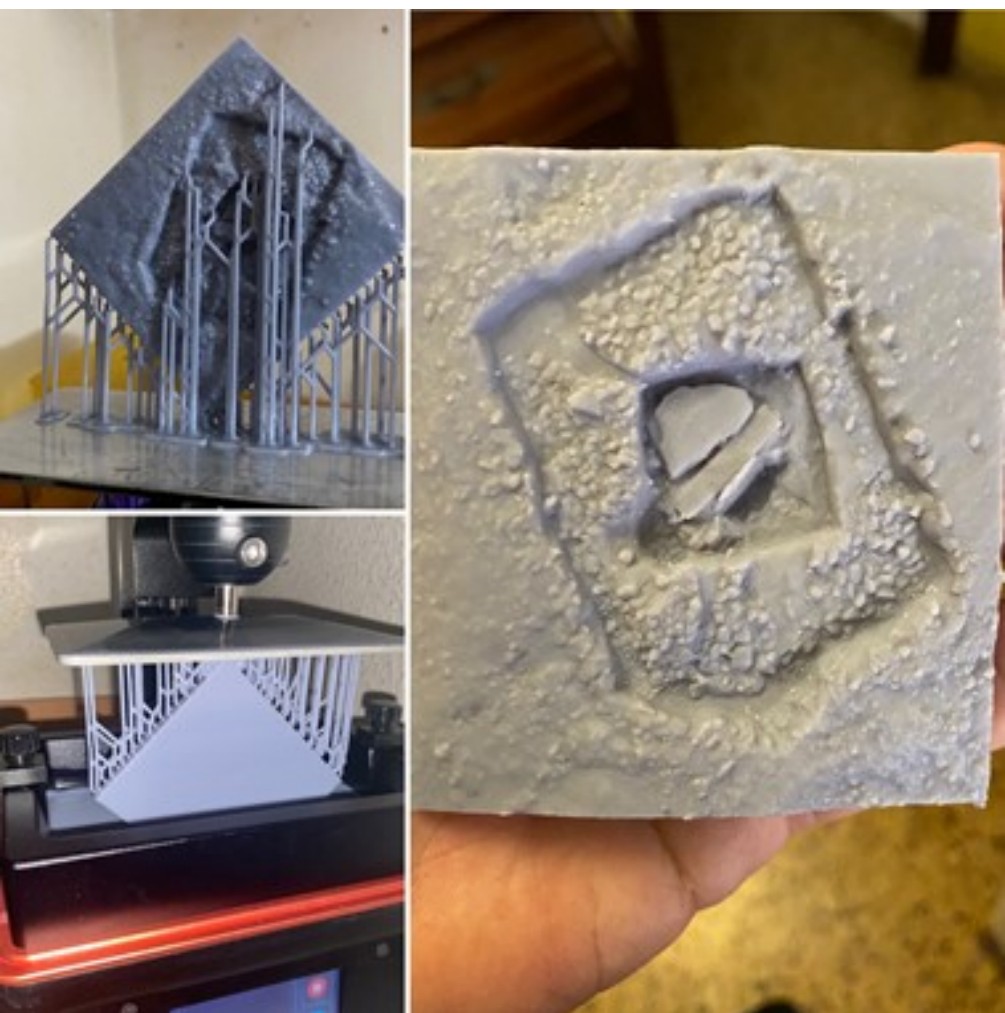

**Figure 16.** Process and result of the last resin test.

For the Puig Rom site, we decided to slice one of the towers discovered during the first excavation campaigns. The photogrammetric model of the said structure was made using the Agisoft Photoscan software. The resulting .obj file was exported to Blender. In the modelling software, the geometry was cleaned, and the possible existing gaps were closed to provide the model with a closed geometry [22]. All of the faces in the model were correctly displayed on the correct sides for a correct texturing (model would serve as the basis for a small digital animation). Unlike the resin-printed models, supports in this case were unnecessary given the piece was flat. In the same way as the Vilanera burial mound, that specific shape was used to facilitate the orientation on and adherence to the print bed.

The result of the first test with FDM printers was a piece measuring 15 × 9 × 5.1 cm, with a layer height of 0.16 mm and a printing duration of approximately 26 h, where we could correctly appreciate the details of the tower construction, especially at its base, where the ashlars stood out above the other elements (Figure 17).

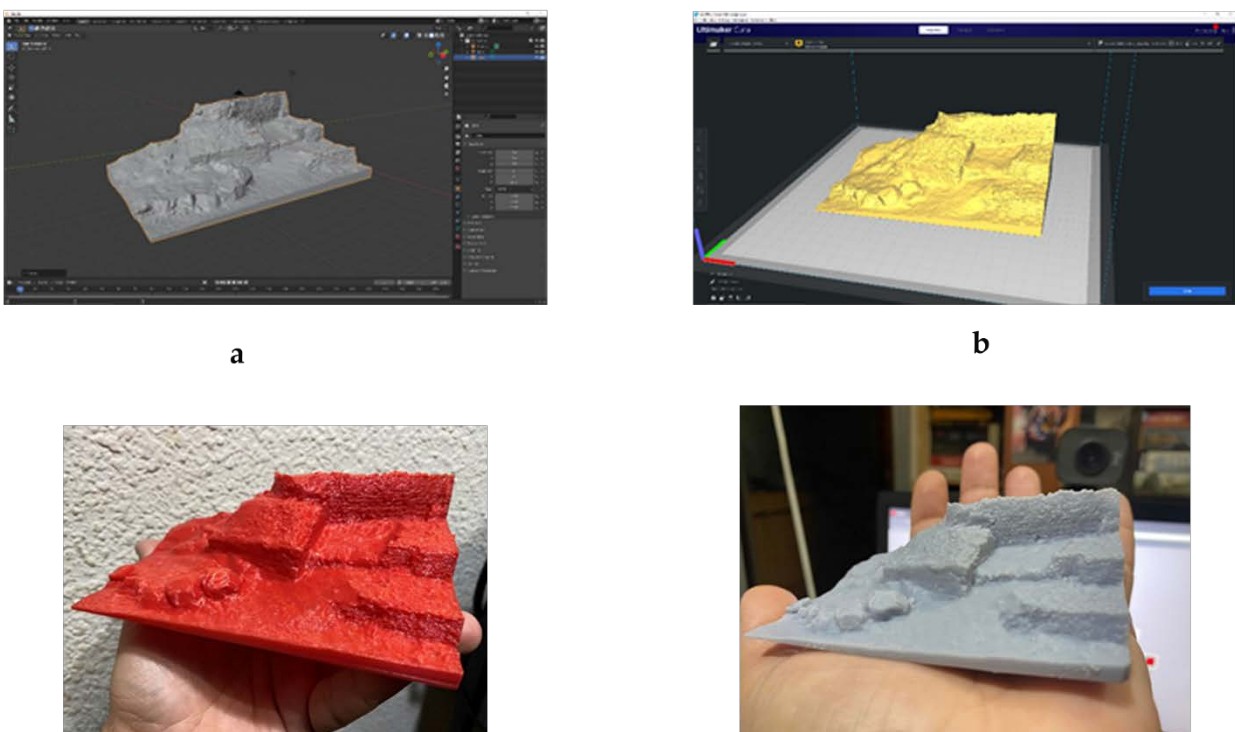

**a**

**b**

**c**

**d**

**Figure 17.** 3D print of the Puig Rom tower. (**a**) .obj file obtained from SFM, cleaned in Blender; (**b**) cleaned .obj sliced in Cura 4.10; (**c**) PLA results from Creality Ender 3; (**d**) resin printout made with an Elegoo Mars.

On the other hand, resin printing was also used to duplicate this model and make the appropriate comparisons of time and quality. For this, the size was modified for hardware reasons, and the print was made with a layer height of 0.05. The final print time for the $10 \times 5 \times 3$ cm reproduction was 11 h. As expected, the details in this print, despite its small size, were far superior to the PLA print.

The 3D prints generated for this article were used for the calibration and fine-tuning tests of the machines with which they were going to work on future projects. The quality/time relationship was sought to be able to carry out these three-dimensional replicas. However, filament printers can be modified to obtain more optimal results by replacing the nozzle with a smaller one (0.2 mm), and also reducing the layer height value to a more optimal 0.08 mm. Anything less than that would have a negative impact on our hardware, as printer engines are tuned to a full cycle number and anything other than what are called 'magic numbers' [23] (0.08, 0.12, 0.16 mm, and any variation in a factor of 0.04 mm) can cause the z axis, the one in charge of lifting the extrusion motor, to suffer problems and this will generate problems in the layers. On the other hand, for LCD printers, the value of the layer height can be significantly reduced, reaching 0.01 mm, but this would considerably increase the printing time, since the layers would be more numerous, and this would imply greater wear on our screen.

## 4. Discussion and Conclusions

First, it should be noted, from a heritage point of view, that RPA contribute to the elaboration of a geometric documentation and is underlined in all the resolutions, letters, and declarations made since the Venice Charter of 1964 for the purposes of conservation, management, evaluation of structural conditions, archive, inventory, publication, and dissemination [24].

Second, and as mentioned by Poirier et al. [1], is the "democratisation" of RPA, which form one of the essential parts of the geometric documentation process through the use

of cameras, and through photogrammetry programs. However, as has been seen, this is also true for low-cost multispectral cameras and remote sensing tasks that are effective in providing results and to which it is possible to apply the same procedures and analytics as those applied to satellite images. The relative low cost of both drones and cameras, as well as the lower cost of the components to geolocate the images obtained with centimetre precision, are aspects that are resulting in the quantitative and qualitative increase in research and in the detection and safeguarding of heritage. The results obtained in these analyses, and following the methodology proposed by Casana et al. [25], allow us to approach a precise detection that must be reflected in subsequent excavation projects at certain points. For the moment, it has been possible to reliably detect traces of previous excavations in other sites analysed outside of this study, such as Ciutadella de Roses (Girona), and which will see the light of day in future publications. That is why using a precise methodology in terms of flight tracing and the ideal configurations of our sensors will allow us to be more precise and concise in the non-invasive detection that is intended to be carried out.

The true purpose of all non-invasive prospecting is to be able to demonstrate, through the execution of a programmed excavation, that those anomalies and data that are suspected to belong to archaeological elements are actually located in the areas that are analysed. It is for this reason that, in order to validate the real effectiveness of the approaches carried out by means of the analyses in the deposits presented, it seems necessary either to carry out small excavations or surveys that provide us with truthful information, or to pass to an intermediate geophysical prospecting that contributes more arguments for the excavation. On the other hand, the terrestrial surveys combined with aerial ones give us even higher rates in terms of the possibility that these structures really are there, or if not, the remains of what had once existed.

The use of RPA in non-invasive prospecting in archaeology constitutes, as Poirier et al. [1] said, a step forward in the possibility of carrying out flights at the most appropriate times for the detection of archaeological traces, contributing to effective diagnosis management. At this time, as has been seen, it is already possible to work with multispectral cameras applying the same methods and techniques that we have used up to now in multispectral images obtained from satellites. Moreover, it is possible to obtain equivalent results with the advantage of working at centimetric resolutions, and more importantly, non-dependence on the periodicity in which these images are made available to the users. Now we are the ones who decide when we take the measurements, and under what conditions. For this, it is important to consider the current legislation in the areas to be flown with regard to the European regulations for the use of RPA, which come into force from January 2022. The relatively moderate cost of the materials purchased leads us to consider data collection with RPA and the combination of multispectral and thermographic cameras as a true methodology of use.

This consideration is a determining factor on many occasions, since the tight budget that archaeological activities usually have in many of the projects carried out means that the choice of the team is an important point, as in 2019, Murtiyoso, Grussenmeyer, and Suwardhi [26] proposed a low-cost flow of choice in both hardware and software to be used for processing the data obtained. To this day, with a small variation in the models of aircraft available, it is still a valid and interesting starting point to be able to decide on the actions.

One of the elements that we proposed in the analytics was to carry out a classification of the dense points obtained through the Agisoft Metashape photogrammetry program. We suggested applying the functions of the RVT program on the DTM resulting from only working with the points classified as ground by the program. We have seen that the results can be equally interesting, considering that we work with 3D models in which the vegetation does not present high densities, which is impossible to filter unless we find point clouds resulting from a LiDAR exploration.

Another question not mentioned comes in reference to the volume of information with which we are manoeuvring. The level of detail and precision that we agreed to give to

the project, the scale of work, the various bands and results of indices and treatments of the Digital Terrain Model, and all of this information that has to be analysed, means large volumes of data; therefore, management must not only be efficient, but requires storage media and also quick access so that the execution times of photogrammetry programs or processing of the resulting images, such as RVT, are not penalised. For example, a QGIS project with the information in layers of all the sites treated in this work was blocked by volume in the "A Coruña" version, and nevertheless the improvements of the last one, "Hannover", fortunately allowed the project file to be raised again and solve the problem.

The third aspect that we would like to highlight in relation to the other two mentioned above is the dissemination and socialisation of the results obtained from the use of RPA. The created 3D models contribute, through virtual reconstructions, to the knowledge and dissemination of the archaeological heritage through virtual manipulation. However, the physical reproductions of the minimum detail of archaeological remains through 3D impressions contribute not only to the understanding and comprehension of the volume and structure of the heritage remains, but also to include other groups such as the blind or age groups whose members prefer physical touch over virtual representation.

Another important point to be addressed in this discussion is the "democratisation" of RPA. Making the possibility of acquiring and piloting an aircraft with certain characteristics available to everyone has its counterpart, since it means that not everyone can assume the role of planning and piloting a flight. Not only is the qualification and compliance with the legislation necessary, but it is also necessary to have knowledge of flight strategies depending on what functionality we want to extract from the data obtained, which only a long period of experience can provide. A bad strategy can lead to results that are neither optimal nor desired, which in the worst case can mean the non-repetition of the flight and therefore result in costs.

One of the possibilities offered by virtual archaeology is the implementation of different proposals to offer a musealisation of the sites. It can generate infographic material that helps the general public understand what is before their eyes. In addition to this is the possibility of proposing reconstructive hypotheses using, for example, the letter of Reconstructive Units proposed by the University of Alicante [27], which is a material that we must incorporate in an obligatory way if we want to contribute the bonus of scientificity that requires us to capture our virtual hypotheses in graphic format and that helps to maintain the principles of Seville, one of the last great documents for the correct use of virtual archaeology in all its aspects.

Closing the circle, we can identify the possibilities of 3D printing. Putting back, in a tangible way, our archaeological hypotheses that have been digitised from real physical remains opens up the possibility of generating suitable content so that groups with sensory difficulties can approach the world of archaeology, and learn about pieces and architectural elements that otherwise could be much more complicated. We mainly talk about those with visual impairments. That is why, observing the progress of 3D printing techniques and the volume of machines whose price is becoming more affordable, we can begin to propose the creation of typological catalogues based on the geometric documentation of heritage elements—from full-scale replicas to scaled pieces, but with a very high detail that allows us to observe all of the imperfections and those smaller elements that are important for understanding the piece. This leads us to propose the possibility of developing Labcase, that is, portable laboratories, in the style of those proposed by Martinez, López, and Santacana [28], that help not only groups at risk of exclusion, but that are also of use in student training processes, including not only the volumetry from the plans of the archaeological structures, but also the construction techniques used.

On many occasions, archaeological work is like reading a book that destroys the pages as the plot progresses. For this reason, and emphasising the previous paragraph, poor planning when geometrically documenting a site can lead to the inability to reconstruct those past pages. That is why optimal planning, and the necessary knowledge, will make the flight—and therefore the work—increase its degree of reliability by a high percent-

age to be able to analyse any element that interests us afterwards. The importance of three-dimensional documentation in current times allows us not only to approach this information from an optimal perspective that is impossible to do from the usual 2D images, but also the correct treatment, storage, and knowledge by the work team can make us go back to a previous phase with all the complete elements to analyse their evolution and make the necessary proposals that lead us to a complete understanding of the heritage element we are studying.

Therefore, it is necessary that we make a subsection regarding the differences in the documentation of patrimonial elements. Although the technique and methodology are the same, their purpose may not be, and for this reason, we must understand geometric documentation as a means of obtaining results, and not a purpose. Despite being 2021, there is still a feeling among the general public that the geometric documentation of heritage is used to make 3D models that are exhibited in virtual galleries, on museum websites, or on any other dissemination platforms. However, part of the importance that this entails is omitted, and that is the possibility of non-invasive study, of being able to facilitate the dissemination of material between researchers quickly and efficiently. Three-dimensional material has the peculiarity that its language is digital, a binary system of 0 and 1, that can be interpreted by teams from different locations without the obligation to use a common language that is understandable by both parties. That is why it is essential that we advance towards a standardisation of geometric documentation techniques, because the result can be understandable and analysed as an autonomous entity. For these documents to have a valid and scientifically precise support, it is necessary that the methodology be as uniform as possible to detect possible errors, propose solutions, and, ultimately, move towards a democratisation of knowledge in the field of virtual archaeology or digital heritage. This must start with the public administrations themselves and their refusal to accept documents in a foreign format in terms of two-dimensional material, which on many occasions hinders the exhaustive knowledge of that person who later wants to approach those results to carry out their own research. We used to say that archaeology is like a book that tears off the pages as the plot progresses, but it is turning into an e-book that is in our hands.

**Author Contributions:** Conceptualization, J.I.F. and P.M.M.; methodology, J.I.F. and P.M.M.; software, J.I.F. and P.M.M.; validation, J.I.F., R.C., D.C., A.C. and E.S.; investigation, J.I.F., R.C. and E.S.; resources, J.I.F. and R.C.; data curation, J.I.F. and P.M.M.; writing—original draft preparation, J.I.F. and P.M.M.; writing—review and editing, J.I.F. and P.M.M.; supervision, R.C., D.C. and E.S.; project administration, J.I.F. All authors have read and agreed to the published version of the manuscript.

**Funding:** The present work was partially funded. Some of the studies in this article are part of projects financed by Generalitad de Catalunya. For the Vilanera site, the results adhere to the project "El Neolític Antic i Mitjà a la desembocadura del Ter actuacions arqueològiques d'època Prehistòrica a Vilanera/Empúries i el seu entorn". CLT009/18/00015. For the results of the studies at the Puig Rom site, please refer to the project "El poblament visigòtic a Roses: del Puig Rom a l'assentament del Conjunt de la Ciutadella. (Roses, Spain)" 437 K117.

**Acknowledgments:** The sad death of Mercé Roca in December 2014 cut short the possibility of continuing the work presented here at the Cosa site. Without her intervention and support, completely facilitating our work, it would not have been possible to have started this long process of experimentation of all the possibilities that UAVs offer us and that we have presented in this work. We would also like to thank Rosa Cuesta Moratinos, and the management of the Research Team of the Archaeological Site Colonia Clunia Sulpicia, for their welcome and facilities provided during our stay in July 2021.

**Conflicts of Interest:** The authors declare no conflict of interest.

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
