# Peer review of "Examples and Results of Aerial Photogrammetry in Archeology with UAV: Geometric Documentation, High Resolution Multispectral Analysis, Models and 3D Printing"

_drones, doi:10.3390/drones6030059_

Round 1

Reviewer 1 Report

This paper considers the use of UAVs in archaeology explicitly, with some examples and results of aerial photogrammetry, multispectral analysis, and 3D applications, in order to set up the necessity of drone use in collecting geospatial data (and not only) in the field of archaeological works.

The subject addressed in this article is appropriate for the journal aims.

The paper reads nicely, and the work is exciting and well-motivated. I enjoyed reading the work, and the authors have done an excellent job in this study.

Recommendations for improving the manuscript:

  • It is mandatory that the authors include in the used references the works of Professor Stefano Campana (University of Sienna, Italy). He specializes in landscape archaeology, remote sensing and archaeological methodology for research, recording and conservation. Professor Campana's work will help the authors to refine their present study. Some of these works can be found here: https://www.researchgate.net/profile/Stefano-Campana-2/research.
  • Also, I recommend the authors to use these articles as well:

- K. Themistocleousa, C. Danezisa, P. Frattinnib, G. Crostab & A. Valagussa (2018). Best practices for monitoring, mitigation, and preservation of cultural heritage sites affected by geo-hazards: the results of the PROTHEGO project, Sixth International Conf. On Remote Sensing and Geoinformation of the Environment (RSCy2018), PIE Vol. 10773, 107730Z · © 2018 SPIE · CCC code: 0277-786X / 18 / $ 18.

Themistocleous, K., Agapiou, A., Cuca, B. and Hadjimitsis, D.G. (2014). 3D Documentation Of Cultural Heritage Sites. Proc. Progress in Cultural Heritage: Documentation, Preservation, and Protection. 5th International Conference, EuroMed 2014, Limassol, Cyprus, Springer LNCS 8740, EuroMed2014 Conference 2014.

I believe that these titles can help the authors to improve the present study, especially to contextualize the use of drones in the archaeological heritage (and not only).

  • A native English speaker should revise the text.

Author Response

Dear Reviewer,

Thank you very much for the kind words, we have tried to correct the translation errors in the document as well as to add some of the suggested publications by Stefano Campana and Themistocleous. We are aware of their works as they have more weight in other publications we are preparing and will be widely cited. We thought that for this article it was perhaps not necessary to add more, given the specific nature of the use of RPA in our sites. In total we have updated with 3 new references and revised the order of appearance.

Many thanks again.

Kind regards,

The authors.

Reviewer 2 Report

The document can be improved. I place below my suggestions:

Line 26 define NGB acronym
Line 61 erase dot (.) before: to analyze.
Line 63-66 please rephrase. Make periods shorter 
line 82 define BE acronym
Line 97 define IP acronym
Figure 4 there is no box drawn in continuous black line
line 132 place in "" the title or maybe in italics to separate it from the rest of the introducing document
line 139 again a misplaced period 
Figure 5 maybe give better resolution images
Line 159 a missing left parenthesis. Missing a continuous line box in black
Line 173 give the exponent symbol for square  
Lines 205 and 206 please make a valid proposal adding verbs to the phrases and not just give titles
Figure 6 add in the legend the Parrot Anafi description
Line 212 replace: horizontally an vertically with the terms along and across flight paths
Line 214 add production after orthophotographs 
Line 253-254 Please rephrase the first period sentence of the paragraph
Line 272 Place period before Actus et al
Line 279 por: what is this
Line 295 Double right parentheses
Line 296 multiple and words does not help reading the text
Line 417 define asl acronym
Figure 12 image c is not in the same scale as the rest. It would be better to be in the same scale in order for the reader to identify the similar features all the images
Line 437 instead of very use high
Line 457 instead of wider impressions use larger printouts
Line 464 speed unit should be given as a ratio and in the denonminator should be a time unit. I assume the correct is 42mm/sec
Lines 495-499 Please make periods shorter. 
Figure 15 The term Impression has a double meaning. Use printout instead.
Lines 504 - 514 Please rephrase the whole paragraph. It seems that most of the text has comes as translation to English using a translator software

Additionally when a 3D printing result is given it is better to define the scale (e.g. 1:200 or 1:100) of the model.
Lines 524-536 Provide a reference of the suggested magic number values even if you get them from a URL or youtube video

Author Response

Dear reviewer,

We have attached in the list below the list of suggested improvements to the article as well as the relevant modifications we have made to the Word document.  We add a new Figure (Figure 17)

  • Line 26 define NGB acronym

Revised. Nir,Green,Blue added to the text when appears for the first time.

  • Line 61 erase dot (.) before: to analyze.

Revised

  • Line 63-66 please rephrase. Make periods shorter 

Revised. The periods are shorter and we re-write some senteces of the paragraph

  • line 82 define BE acronym

We remove BE. BE it’s a Scolarship, in Spain was called Beca Exterior or Beca d’Estudis.

  • Line 97 define IP acronym

IP it’s a Prinicpal Investigator in Spanish. We put PI instead of IP.

  • Figure 4 there is no box drawn in continuous black line.

Correct. It’s a mistake from a previous version of this figure. In the text we’ve change black line for red line and discontinuous.

  • line 132 place in "" the title or maybe in italics to separate it from the rest of the introducing document

We’ve added Italics to the name of the two projects cited in the text.

  • line 139 again a misplaced period 

Revised

  • Figure 5 maybe give better resolution images

A new image was added to modify the Figure 5.

  • Line 159 a missing left parenthesis. Missing a continuous line box in black

Revised

  • Line 173 give the exponent symbol for square  

Revised

  • Lines 205 and 206 please make a valid proposal adding verbs to the phrases and not just give titles.

Correct. Was a mistake. “Treatment and analysis software” it’s a new point. Not a phrase.

  • Figure 6 add in the legend the Parrot Anafi description

Revised

  • Line 212 replace: horizontally an vertically with the terms along and across flight paths

Modified. Along and acrros are more accurate for the Spanish words “ a lo largo y ancho”.

  • Line 214 add production after orthophotographs 

Revised

  • Line 253-254 Please rephrase the first period sentence of the paragraph

Revised. New text added.

  • Line 272 Place period before Actus et al

Revised

  • Line 279 por: what is this

Was a mistake in the translation, we don’t put “by” to indicate what person studied this methodology

  • Line 295 Double right parentheses

Revised

  • Line 296 multiple and words does not help reading the text

Revised

  • Line 417 define asl acronym

Revised. Asl (above sea level) was defined the first time when appears in this sentences

  • Figure 12 image c is not in the same scale as the rest. It would be better to be in the same scale in order for the reader to identify the similar features all the images.

In this case, the Figure 12c, vas a zoom in for a specific section. We indicate that in the legend of the figure. We don’t modified the scale of this figure.

  • Line 437 instead of very use high

Revised

  • Line 457 instead of wider impressions use larger printouts

Revised

  • Line 464 speed unit should be given as a ratio and in the denonminator should be a time unit. I assume the correct is 42mm/sec

Revised

  • Lines 495-499 Please make periods shorter. 

Revised

  • Figure 15 The term Impression has a double meaning. Use printout

Revised

  • Lines 504 - 514 Please rephrase the whole paragraph. It seems that most of the text has comes as translation to English using a translator software

Paragraph revised and re-writed in some points.

Additionally when a 3D printing result is given it is better to define the scale (e.g. 1:200 or 1:100) of the model.

  • Lines 524-536 Provide a reference of the suggested magic number values even if you get them from a URL or youtube video

URL Added to the reference list.
